# Association of *NUDT17* rs9286836 and rs2004659 variants with breast cancer risk in Bangladeshi Women

Md. Shajid Hossain Rafi[1], Md. Shalahuddin Millat[2], Md Abdul Barek[2],
Syed Masudur Rahman Dewan[3], Mohammad Shahriar[1], Zabun Nahar[1],
Mohammad Safiqul Islam[2]*

1 Department of Pharmacy, School of Pharmacy, University of Asia Pacific, Dhaka, Bangladesh, 2 Department of Pharmacy, Faculty of Biological Sciences, Noakhali Science and Technology University, Noakhali, Bangladesh, 3 Department of Pharmacy, School of Life Sciences, United International University, Dhaka, Bangladesh

* research_safiq@yahoo.com, research_safiq@nstu.edu.bd

## Abstract

Breast cancer is a multifaceted illness impacted by genetic factors as well as environmental influences. While variants in the *NUDT17* gene have been associated with cancer biology, their involvement in breast cancer is still inadequately investigated, especially within South Asian populations. This study investigated the association between two *NUDT17* polymorphisms, rs9286836 and rs2004659, and breast cancer risk in Bangladeshi women using a case–control design. A total of 240 breast cancer patients and 240 age-, sex-, and BMI-matched healthy controls were enrolled. Genomic DNA was extracted from blood samples, and genotyping was performed using tetra-primer ARMS-PCR for rs9286836 and PCR–RFLP for rs2004659. The statistical methods employed included chi-square tests for genotype distributions and logistic regression to calculate odds ratios (ORs) with 95% confidence intervals (CIs). Genotyping for NUDT17 rs9286836 was successfully completed for all recruited participants. However, for rs2004659, high-quality genotyping data were available for 204 breast cancer cases and 204 controls after quality control procedures. No significant association was observed between rs9286836 and breast cancer risk across the tested genetic models. In contrast, rs2004659 showed a robust protective association with breast cancer, particularly among heterozygous individuals, and this protective effect was consistently observed across the dominant, over-dominant, and allelic models. Haplotype analysis revealed that the AA haplotype was associated with an increased risk of breast cancer, whereas the AG and GA haplotypes were associated with a reduced risk. Expression analysis further demonstrated elevated *NUDT17* levels in breast cancer tissues and genotype-dependent expression effects for both variants. These findings suggest a protective role of rs2004659 in breast cancer susceptibility and highlight the potential of *NUDT17* polymorphisms as biomarkers in Bangladeshi women.

**Data availability statement:** Yes - all data are fully available without restriction; All relevant data are within the paper and its Supporting Information files.

**Funding:** The study was partially financially supported by the Department of Pharmacy at the University of Asia Pacific. The funders had no role in study design, data collection and analysis, decision to publish, or preparation of the manuscript.

**Competing interests:** The authors have declared that no competing interests exist.

## 1. Introduction

Breast cancer is the most common cancer in women around the world. It is a complex disease with many different symptoms and a complicated cause, making it hard to come up with global prevention and treatment plans [1,2]. In 2022, there were about 2.3 million new cases of breast cancer and about 666,000 deaths from breast cancer around the world. This means that breast cancer made up 23.8% of all cancer cases and 15.4% of all cancer deaths among women [3]. If current trends continue, breast cancer will cause 3.2 million new cases and 1.1 million deaths each year by 2050. This is a 38% and 68% increase in new cases and deaths, respectively [4].

The Global Cancer Observatory (GLOBOCAN) reported in 2022 that Bangladesh had 12,989 new cases of breast cancer, which was 18.0% of all new cancer cases in women. Also, the 5-year prevalence of breast cancer in Bangladesh was 35,269 cases, corresponding to a prevalence rate of 42.4 per 100,000 people [5]. Studies have shown that inadequate knowledge and education about breast cancer among people in Bangladesh contribute to a higher likelihood of the disease [6]. Breast cancer susceptibility is affected by factors such as age, hormone levels, genetic inheritance, and environmental influences, with hereditary predisposition accounting for 5–10% of cases involving genetic variation [7].

Genome-wide association studies (GWAS) have also found several genes related to breast cancer, as well as single-nucleotide polymorphisms (SNPs) that are linked to the disease's onset and progression. These genes include MAP3K1 [8], ERCC2 [9], BARD1 [10], PALB2 [11], and RENT [12]. The NUDT/ NUDIX (Nucleotide Diphosphate Linked to Moiety X) hydrolase superfamily comprises a structurally conserved collection of proteins that catalyze hydrolysis of substrates, including (d)NTPs, oxidized (d)NTPs, polyphosphates, and capped mRNAs, contributing significantly to vital biological processes such as cell proliferation, signal transduction, and homeostasis [13,14]. All members of the NUDT family are distinguished by a highly conserved 23-residue sequence motif known as the Nudix box, and function as housecleaning enzymes [15,16]. The NUDT gene family may play significant roles in cancer proliferation and metastasis [17]. NUDT1 overexpression correlates with disease advancement in oral squamous cell carcinoma [18] and demonstrates reduced overall survival (OS) and progression-free survival (PFS) rates in colorectal cancer [19]. Elevated NUDT10 expression was correlated with advanced tumour progression, increased local invasion, and poorer survival outcomes in gastric cancer patients [20]. A decreased expression of NUDT21 correlates with tumor size, stage, and metastasis, which are related to lowered overall survival and enhanced recurrence-free survival rates in breast cancer patients [21]. Increased expression of NUDT2 in invasive ductal carcinoma is linked to unfavorable clinical outcomes [22]. NUDT5 overexpression correlates with more aggressive breast tumors [23]. These findings indicate that alterations in NUDT-related nucleotide metabolism are relevant to cancer development, including breast cancer. Within this gene family, *NUDT17* has been less extensively studied, particularly in relation to breast cancer. Located on chromosome 1q21.1, *NUDT17* is a protein-coding gene and a close homolog of MTH1, with

an incompletely understood role in cancer [24]. Given the importance of oxidative nucleotide damage in carcinogenesis, variation in *NUDT17* may influence individual differences in breast cancer susceptibility.

Previous research has shown that the rs9286836 and rs2004659 polymorphisms may change how genes are expressed and the risk of cancer in some populations [25]. These variants are in chromosomal areas that could affect transcriptional regulation and change how cells deal with oxidative stress. However, evidence remains limited, and no data are available for South Asian populations.

A recent case-control study of Chinese women found strong links between NUDT17 polymorphisms, such as rs9286836 and rs2004659, and the risk of getting breast cancer [25]. To date, these associations have not been examined in Bangladeshi women. Considering the burden of breast cancer in Bangladesh and the limited availability of population-specific genetic data, examining these variants in this population is of particular interest. Although other NUDT17 variants, such as rs10910830 and rs10910829, have been reported as potential regulatory SNPs, the present study focused on rs9286836 and rs2004659 based on prior evidence of functional relevance, reported associations in other populations, and feasibility within the current study design.

## Materials and methods

### Ethical clearance

The Ethical Review Board at the University of Asia Pacific (UAP) granted approval for the study on October 6, 2024, under reference number UAP/REC/2024/207. Additionally, ethical approval was obtained from the Ethical Committee of the National Institute of Cancer Research and Hospital (NICRH) on March 25, 2023, which is referenced as NICRH/IRB/2023/96. Patient recruitment and blood sample collection were conducted at NICRH under these institutional approvals, with clinical assistance from NICRH healthcare staff, while all laboratory analyses, data analysis, and interpretation were performed by the study investigators.

### Participant group

The study followed a population-based case-control approach, wherein persons diagnosed with breast cancer were designated as cases, and those with neither previous records of breast cancer nor other long-term health conditions were defined as controls. A group of 240 women with confirmed breast cancer diagnosis, regardless of their disease stage, was selected as the case group, along with 240 healthy controls frequency-matched by age, sex, and body mass index (BMI).

Participants for the study were recruited from the NICRH, from where blood samples and clinical information (age, sex, BMI, TNM staging, demographics, and lifestyle factors) were collected with the help of healthcare professionals. Healthy individuals without a history of malignancies were selected from various locations within the Dhaka capital area. Participants with prior cancer diagnoses, chronic inflammatory diseases, or ongoing chemotherapy unrelated to breast cancer were excluded. Participants were recruited from March 28, 2023, to June 25, 2025. Written informed consent was obtained from all participants prior to the collection of samples and data. The research adhered to the Declaration of Helsinki, issued by the World Medical Association, along with its later revisions [26]. All laboratory analyses were performed at the Bio-Technology Laboratory for research within the Department of Pharmacy at the University of Asia Pacific, Dhaka, Bangladesh.

### Collection of blood samples and subsequent DNA extraction

Five milliliters of venous blood specimens have been collected from the patient group and control group in accordance with the blood collection protocol defined by the WHO. These samples were immediately placed in an EDTA tube and preserved in a freezer at −80°C before the DNA extraction. The extraction of genomic DNA from whole blood was performed using the method outlined by Islam et al. [27] with the aid of a DNA extraction kit supplied by Favorgen (Taiwan). A micro-volume spectrophotometer (Optima, Japan) was utilized to evaluate the concentration as well as the purity of the

DNA sample. The purity of the extracted DNA was evaluated by measuring the absorbance ratio A260:A280, with samples showing a ratio between 1.7 and 1.9 considered pure.

## Primer design

There are various online software tools accessible for primer design. For rs9286836, we utilized Primer1 software (https://primer1.soton.ac.uk/primer1.html) to design the four necessary primers, which are listed in Table 1. The two primers required for rs2004659 were designed using Primer Blast, adhering to the MIQE guidelines [28], as outlined in Table 1.

## Genotyping of rs9286836 and rs2004659

Different genotyping methods were selected based on SNP-specific sequence characteristics and methodological feasibility; rs9286836 lacks a suitable restriction enzyme recognition site for PCR–RFLP and was therefore genotyped using tetra-primer ARMS-PCR, whereas rs2004659 contains an appropriate restriction site enabling accurate PCR–RFLP analysis. The tetra-primer ARMS-PCR technique was employed in the genotyping of rs9286836, as outlined by Aziz et al. [29]. PCR amplification was done in an 11 µl reaction mixture that included 10 µL of the working PCR mix, which consisted of Premix Taq™ (Takara, Japan) and four specifically designed complementary primers at appropriate concentrations. The initial optimization explored a range of annealing temperatures between 60°C and 65°C, with adjusted primer concentrations and optimal amplification occurred at 63°C. The final cycling conditions are detailed in Table 1. The amplified products were subjected to separation via 3% agarose gel electrophoresis and subsequently stained with ethidium bromide, and a DNA ladder of 100 bp was used for visualization under UV light. The anticipated fragment sizes are listed in Table 1.

For analysis of rs2004659, genotyping was carried out using the PCR-RFLP approach. A PCR reaction mixture was prepared with Premix Taq™ and a pair of specifically designed forward

and reverse primers at optimized concentrations. 20 µl of this mixture was mixed with 1µl of template DNA, giving a 21µl reaction volume. Table 1 shows the amplification conditions and the sizes of the products that are predicted. Amplified products were first evaluated on a 1% agarose gel to ensure the target fragments. The verified PCR products were subsequently digested with speific restriction endonucleases, as shown in Table 1, to make the RFLP patterns. We employed a 100 bp DNA ladder to estimate the sizes of the digested DNA fragments as they were electrophoresed on a 2% agarose gel. For genotyping quality control, approximately 10% of randomly selected samples were re-genotyped, yielding 100% concordance.

**Table 1. The primer sequences of *NUDT17* rs9286836 and *NUDT17* rs2004659 polymorphisms PCR conditions and related parameters.**

| SNP | Primers (5′-3′) | PCR Conditions | No. Cycles | Amplified band (bp) | Restriction enzyme | Digestion Conditions | Digested fragments size (bp) |
|---|---|---|---|---|---|---|---|
| rs9286836 | FI:5`-ACCATCTTGGGTGTGGGCTGAGGGCGA-3` RI:5`-CCTGACTCTTCCAAGTCCTTTTGCTTGACC-3` FO:5`-ACCACCACCATTCACCTACTGGCCCCAC-3` RO: 5`-TCACCTCCCATAAACCCAGAGGGACCCA-3` | 94 °C 5 min | 1 | AA: 190,301 AG:168, 190, 301 GG: 168, 301 | – | – | – |
| | | 94 °C 1 min | 35 | | | | |
| | | 63 °C 45 s | | | | | |
| | | 72 °C 45 s | | | | | |
| | | 72 °C 7 min | 1 | | | | |
| rs2004659 | FP: 5`-ATAACCCTCACCCATGTCCC-3` RP:5`-TAGTAGAGACGGGGTTTCGC-3` | 94 °C 5 min | 1 | 238 | Tru9I (MseI) | t 65°C 16 hrs | AA: 89.149 AG: 89, 149.238 GG: 238 |
| | | 94 °C 30 s | 35 | | | | |
| | | 60 °C 30 s | | | | | |
| | | 72 °C 30 s | | | | | |
| | | 72 °C 7 min | 1 | | | | |

### In silico gene expression

In silico expression analyses were performed using publicly available datasets. Expression differences between breast cancer and normal tissues were examined with OncoDB (http://oncodb.org), while genotype-specific expression effects for rs9286836 and rs2004659 were retrieved from the GTEx portal (https://gtexportal.org). Primary tumor tissue expression data were not available from the study cohort; therefore, publicly validated datasets were used. Reported *p*-values were directly obtained from these databases.

### Statistical evaluation

Statistical analysis was done using the SPSS (version 25.0) software program and MedCalc (version 23.1.6). Demographic data variations between patients and controls were evaluated using χ2-tests and two-sided unpaired t-tests. Odds ratios (ORs) with 95% confidence intervals (CIs) were calculated using logistic regression analysis to investigate the relationship between genotypes and breast cancer risk, with the wild-type genotype serving as the reference category. All statistical analyses were performed using the maximum number of successfully genotyped samples for each SNP; accordingly, rs9286836 analyses included 240 cases and 240 controls, whereas rs2004659 analyses were conducted using 204 cases and 204 controls. An odds ratio exceeding 1.00 indicates a positive correlation with risk, while values less than 1.00 indicate a protective effect. Hardy–Weinberg equilibrium (HWE) was assessed in the control group using a chi-square goodness-of-fit test.

Additionally, the online tool SHEsis was utilized for conducting haplotype analysis (http://analysis.bio-x.cn/) to estimate haplotype frequencies and assess their association with breast cancer risk. All statistical analyses were performed using a two-tailed test, with p < 0.05 considered statistically relevant. Bonferroni correction was pre-specified to control for multiple testing across six genetic models, with an adjusted significance threshold of p < 0.0083. Crude logistic regression models were primarily reported due to collinearity among reproductive and lifestyle variables; this is acknowledged as a limitation.

An a priori sample size estimation was performed in G*Power (v3.1.9.7) using a chi-square test of independence under a dominant model (AG + GG vs AA; df = 1, α = 0.05, power = 0.80). The effect size (Cohen's w = 0.14) was estimated from pilot genotype frequencies. The required total sample size was N = 400 (200 cases and 200 controls). The final analysis for rs2004659 included 408 participants (204 breast cancer cases and 204 controls), which satisfied the predefined statistical power requirement. In contrast, the analysis of rs9286836 was performed using the entire recruited cohort, comprising 240 cases and 240 controlsde.

## 3. Results

### Sociodemographic profile of the study participants

The fundamental sociodemographic and reproductive characteristics of the study subjects, encompassing aspects such as religion, body mass index (BMI), marital status, reproductive history, and lifestyle-related factors, are detailed in Table 2 and S1 Table.

The predominant demographic in both the cases and controls was Muslim, comprising 90.83% and 97.08%, respectively, followed by a small fraction of Hindus and a minimal number of Christians. The control group had a higher frequency of underweight individuals (12.50%) than the case group (8.33%); the majority of participants in both groups exhibited a normal BMI. The majority of individuals in both categories were aged above 40. While 26.25% of the control group were single, all breast cancer patients were married. While the majority of control individuals (66.28%) had two or fewer children, almost half of the breast cancer patients (47.83%) had more than two children. Comparable distributions were observed amongst cases and controls for age at first menstruation, onset of menopause, and age at first childbirth.

The length of breastfeeding showed considerable variation (p < 0.001), with 62.72% of patients breastfeeding for more than two years compared to 35.41% of controls. Breast cancer patients were more likely to have more than two children

Table 2. Socio-demographic information of breast cancer patients and controls.

| Types of Features | Case, n (%) | Control, n (%) | P Value |
|---|---|---|---|
| BMI | | | |
| Mean (SD) | 24.50 | 24.34 | 0.557 |
| Underweight (<18.5) | 20 (8.33) | 30 (12.50) | 0.436 |
| Normal (18.5–24.99) | 118 (49.17) | 117 (48.75) | |
| Overweight (25–29.99) | 76 (31.67) | 66 (27.50) | |
| Obese (≥30) | 26 (10.83) | 27 (11.25) | |
| | Marital Status | | |
| Married | 240 (100) | 177 (73.75) | – |
| Unmarried | 0 (0) | 63 (26.25) | |
| Age (Years) | | | |
| Average (SD) | 42.9(7.03) | 41.77(9.58) | 0.126 |
| ≤40 | 92 (38.33) | 100 (41.67) | 0.456 |
| >40 | 148 (61.67) | 140 (58.33) | |
| Number of Children | | | |
| 1-2 | 120 (52.17) | 114 (66.28) | **0.019** |
| >2 | 110 (47.83) | 58 (33.72) | |
| Menstrual Cycle Starting Age (Years) | | | |
| ≤13 | 166 (69.17%) | 194 (80.83) | **0.003** |
| >13 | 74 (30.83%) | 46 (19.17) | |
| Menstrual Cycle Stopping Age (Years) | | | |
| ≤45 | 120 (66.67) | 15 (41.67) | **0.006** |
| >45 | 60 (33.33) | 21 (58.33) | |
| First Child Conceived (year) | | | |
| ≤18 | 100 (43.48) | 80 (46.51) | 0.545 |
| >18 | 130 (56.52) | 92 (53.49) | |
| Breastfeeding Period (Years) | | | |
| ≤2 | 85 (37.28) | 111 (64.53) | <0.001 |
| ≥2 | 143 (62.72) | 61 (35.47) | |
| History of Taking Contraceptive Pills | | | |
| Yes | 148 (61.67) | 118 (49.17) | 0.006 |
| No | 92 (38.33) | 122 (50.83) | |

(47.83%) compared to controls (33.72%; p = 0.019). In contrast, early menarche (≤13 years) was less frequent among patients (69.17%) than controls (80.83%; p = 0.003). The utilization of contraceptive pills was more prevalent among breast cancer patients (61.67%) than among controls (49.17%; p = 0.006). The groups exhibited no significant variations concerning BMI, age, first child conceived age, age of first delivery, gap between children or family history of cancer (p > 0.05). Moreover, a negligible number of individuals from each cohort reported utilizing postmenopausal hormone therapy or smoking.

## Clinicopathological attributes of breast cancer patients

Table 3 presents the clinicopathological characteristics of the breast cancer patients included in this study. The predominant histological subtype was invasive ductal carcinoma (IDC, 77.08%), followed by invasive lobular carcinoma (ILC, 10.83%), ductal carcinoma in situ (DCIS, 7.92%), metastatic ductal carcinoma (MDC, 3.33%), and medullary carcinoma (MC, 0.83%). The most frequent tumor grades were Grade 2 (50.0%) and Grade 3 (47.92%), with a few cases of Grade

**Table 3. Clinicopathological Characteristics of Breast Cancer Patients.**

| Types of Breast Cancer | | Tumor Stage | | Tumor Grade | | Tumor Size | | Receptor status | |
|---|---|---|---|---|---|---|---|---|---|
| Ductal Carcinoma In Situ (DCIS) | 19 (7.92%) | 0 | 19 (7.92) | Grade 1 | 5 (2.08%) | T1 | 86 (35.83%) | Triple Negative | 12 (5.0%) |
| Invasive Ductal Carcinoma (IDC) | 185(77.08%) | IA | 59 (24.58%) | Grade 2 | 120 (50.0%) | T2 | 66 (27.50%) | Single Positive | 120 (50.0%) |
| Invasive Lobular Carcinoma (ILC) | 26 (10.83%) | IB | 10 (4.17%) | Grade 3 | 115 (47.92%) | T3 | 61 (25.42%) | Double Positive | 93 (38.75%) |
| Medullary Carcinoma (MC) | 2 (0.83%) | IIA | 45 (18.75%) | | | T4 | 27 (11.25%) | Triple Positive | 15 (6.25) |
| Metastatic Ductal Carcinoma (MDC) | 8 (3.33%) | IIB | 30 (12.50%) | **Lymph Node Status** | | **Distant Metastasis** | | **Nodal Status** | |
| | | IIIA | 20 (8.33%) | Positive | 137 (57.08%) | M0 | 201 (83.75%) | N0 | 90 (37.50%) |
| | | IIIB | 19 (7.92%) | Negative | 103 (42.92%) | M1 | 39 (16.25%) | N1 | 58 (24.17%) |
| | | IVA | 6 (2.50%) | | | | | N2 | 37 (15.42%) |
| | | IVB | 32 (13.33%) | | | | | N3 | 42 (17.08%) |
| | | | | | | | | Nx | 14 (5.83%) |

1 (2.08%). Tumor staging revealed the highest prevalence of Stage IA (24.58%), followed by Stage IIA (18.75%), Stage IIB (12.50%), Stage IVB (13.33%), and Stage IIIA (8.33%). Regarding tumor size, T1 (35.83%) and T2 (27.50%) tumors were the most common, followed by T3 (25.42%) and T4 (11.25%). Lymph node analysis showed positive involvement in 57.08% of patients and negative in 42.92%. Among the nodal categories, N0 (37.50%) was the most frequent, followed by N1 (24.17%), N2 (15.42%), N3 (17.08%), and Nx (5.83%). Distant metastasis was absent (M0) in 83.75% of patients and present (M1) in 16.25%. Based on receptor status, single-positive tumors (50.0%) were most common, followed by double-positive (38.75%), triple-negative (5.0%), and triple-positive (6.25%) cases.

## Association of Breast Cancer with *NUDT17* rs9286836 Polymorphism

Genotype and allele distributions of NUDT17 rs9286836 were analyzed in 240 breast cancer cases and 240 healthy controls. Gel electrophoresis images for rs9286836 are provided in Supplementary **S1 Fig**. The genotypes and allele frequencies of the *NUDT17* rs9286836 polymorphism among cases and healthy controls are shown in Table 4. In this study, 64.17% of cases and 58.75% of controls exhibited the AA genotype, whereas the AG genotype was detected in 30.00% of cases and 37.92% of controls, and the GG genotype was identified in 5.83% of cases and 3.33% of controls. The genotype distributions in both cases (p = 0.161) and controls (p = 0.143) were in Hardy–Weinberg equilibrium, indicating no deviation from expected genotype frequencies.

Various genetic models were applied to assess the association between rs9286836 and breast cancer risk (Table 4). Although some models showed trends toward either decreased or increased risk, none of these associations reached statistical significance (all p > 0.05). Specifically, the additive, dominant, over-dominant, and allelic models showed nonsignificant trends toward reduced risk, whereas the additive-2 (GG vs. AA) and recessive models showed nonsignificant trends toward increased risk. Overall, rs9286836 was not significantly associated with breast cancer risk in this population.

## Association of breast cancer with *NUDT17* rs2004659 polymorphism

Gel electrophoresis images for rs2004659 are provided in Supplementary **S2 Fig**. Genotype and allele distributions of NUDT17 rs2004659were analyzed in 204 breast cancer cases and 204 healthy controls. Genotype and allele distributions for rs2004659 are shown in Table 5. The AA genotype was more frequent among cases than controls (61.76% vs. 49.02%), whereas the AG genotype was more frequent in controls than in cases (45.10% vs. 32.35%). The GG genotype occurred at the same frequency in both groups (5.88%). Genotype distributions in cases (p = 0.398) and controls (p = 0.122) were in Hardy–Weinberg equilibrium.

**Table 4. Association of *NUDT17* rs9286836 with breast cancer in Bangladeshi women.**

| Genetic models | Genotype | Cases (N = 240) (%) | Controls (N = 240) (%) | Crude analysis | |
|---|---|---|---|---|---|
| | | | | OR (95% CI) | *p*-value |
| Additive model 1 (AG vs. AA) | AA | 154 (64.17) | 141 (58.75) | 1 | |
| | AG | 72 (30) | 91 (37.92) | 0.72 (0.49 to 1.06) | 0.100 |
| Additive model 2 (GG vs. AA) | AA | 154 (64.17) | 141 (58.75) | 1 | |
| | GG | 14 (5.83) | 8 (3.33) | 1.60 (0.65 to 3.93) | 0.304 |
| Dominant model (AG + GG vs. AA) | AA | 154 (64.17) | 141 (58.75) | 1 | |
| | AG + GG | 86 (35.83) | 99 (41.25) | 0.80 (0.55 to 1.15) | 0.223 |
| Recessive model (GG vs. AA + AG) | AA + AG | 226 (94.17) | 232 (96.67) | 1 | |
| | GG | 14 (5.83) | 8 (3.33) | 1.80 (0.74 to 4.36) | 0.196 |
| Over-dominant model (AG vs. AA + GG) | AA + GG | 168 (70) | 149 (62.08) | 1 | |
| | AG | 72 (30) | 91 (37.92) | 0.70 (0.48 to 1.03) | 0.068 |
| Allelic model (A vs. G) | A | 380 (79.17) | 373 (77.71) | 1 | |
| | G | 100 (20.83) | 107 (22.29) | 0.92 (0.67 to 1.25) | 0.583 |

**Table 5. Association of *NUDT17* rs2004659 with breast cancer in Bangladeshi women.**

| Genetic models | Genotype | Cases (N = 204) (%) | Controls (N = 204) (%) | Crude analysis | |
|---|---|---|---|---|---|
| | | | | OR (95% CI) | *p*-value |
| Additive model 1 (AG vs. AA) | AA | 126 (61.76) | 100 (49.02) | 1 | |
| | AG | 66 (32.35) | 92 (45.10) | 0.57 (0.38 to 0.86) | **0.007** |
| Additive model 2 (GG vs. AA) | AA | 126 (61.76) | 100 (49.02) | 1 | |
| | GG | 12 (5.88) | 12 (5.88) | 0.79 (0.34 to 1.84) | 0.591 |
| Dominant model (AG + GG vs. AA) | AA | 126 (61.76) | 100 (49.02) | 1 | |
| | AG + GG | 78 (38.24) | 104 (50.98) | 0.59 (0.40 to 0.88) | **0.0098** |
| Recessive model (GG vs. AA + AG) | AA + AG | 192 (94.12) | 192 (94.12) | 1 | |
| | GG | 12 (5.88) | 12 (5.88) | 1.00 (0.44 to 2.28) | 1.000 |
| Over-dominant model (AG vs. AA + GG) | AA + GG | 138 (67.65) | 112 (54.90) | 1 | |
| | AG | 66 (32.35) | 92 (45.10) | 0.58 (0.39 to 0.87) | **0.009** |
| Allelic model (A vs. G) | A | 318 (77.94) | 292 (71.57) | 1 | |
| | G | 90 (22.06) | 116 (28.43) | 0.71 (0.52 to 0.98) | **0.037** |

Unlike rs9286836, rs2004659 showed statistically significant associations with breast cancer risk. In the additive model (AG vs. AA), the AG genotype was associated with a significantly reduced risk (OR = 0.57, p = 0.007). Similarly, the dominant (AG + GG vs. AA) and over-dominant (AG vs. AA + GG) models demonstrated significant protective effects (p = 0.0098 and p = 0.009, respectively). The allelic model also showed a significant reduction in risk for the G allele (p = 0.037).

After Bonferroni correction, the additive model (AG vs. AA) remained statistically significant, confirming a robust protective association of rs2004659 with breast cancer risk, whereas the other models became borderline or nonsignificant.

### Haplotype analysis of *NUDT17* rs9286836 and rs2004659

The haplotype analysis of rs9286836 and rs2004659 demonstrated notable differences in distribution between breast cancer cases and healthy controls (Table 6). The AA haplotype was the most frequent, occurring in 74.7% of cases compared to 63.8% of controls, and was significantly associated with increased breast cancer risk (OR = 1.67, 95% CI = 1.24–2.26,

**Table 6. Haplotype frequencies of *NUDT17* rs9286836 and rs2004659 polymorphisms and their association with the risk of breast cancer.**

| Genotype | Frequency | | Odds Ratio (95% CI) | p |
|---|---|---|---|---|
| | Cases | Controls | | |
| AA | 0.747 | 0.638 | 1.67 (1.24–2.26) | 0.001 |
| AG | 0.028 | 0.132 | 0.19 (0.10–0.36) | <0.001 |
| GA | 0.033 | 0.078 | 0.40 (0.21–0.77) | 0.005 |
| GG | 0.193 | 0.153 | 1.33(0.92–1.91) | 0.128 |

p = 0.001). In contrast, the AG haplotype appeared at a lower frequency among cases (2.8%) than controls (13.2%), indicating a statistically significant protective effect (OR = 0.19, 95% CI = 0.10–0.36, p < 0.001).

The GA haplotype (3.3% in cases vs. 7.8% in controls) suggested a statistically significant protective effect (OR = 0.40, 95% CI = 0.21–0.77, p = 0.005). In contrast, the GG haplotype exhibited comparable frequencies in cases (19.3%) and controls (15.3%), with no meaningful association (OR = 1.33, 95% CI = 0.92–1.91, p = 0.128).

Linkage disequilibrium (LD) analysis revealed a moderate correlation between the two loci, with D′ = 0.688 and r² = 0.414, indicating partial but not complete non-random association. The LD plot presented in Fig 1 illustrates the haplotype block structure of rs9286836 and rs2004659, confirming moderate linkage between these variants (Table 6, Fig 1)

### *NUDT17* expression analysis

*NUDT17* expression analysis was performed using publicly available datasets. Expression profiling from OncoDB (Fig 2a) showed significantly higher *NUDT17* levels in breast cancer tissues compared with normal samples (p = $1.1 \times 10^{-6}$). Genotype-specific analysis from GTEx revealed that rs2004659 was associated with reduced *NUDT17* expression in AG and GG carriers compared with AA homozygotes (p = $1.5 \times 10^{-13}$; Fig 2b) in the breast mammary tissues. Similarly, rs9286836 showed a strong effect on *NUDT17* expression across genotypes (p = $2.73 \times 10^{-14}$; Fig 2c).

## Discussion

This investigation was carried out in Bangladesh to examine the potential effect of *NUDT17* gene polymorphisms on breast cancer susceptibility. Previous studies on the *NUDT17* rs9286836 and rs2004659 polymorphisms suggested a possible correlation between these genetic variants and cancer susceptibility [25]. We initially assumed that these polymorphisms could be relevant to breast cancer progression in the Bangladeshi population. In a Chinese cohort, Han et al. (2024) reported that the G allele of rs2004659 was linked to a lower risk of breast cancer in allelic analysis. A comparable protective pattern for this allele was evident in our Bangladeshi cohort [25]. While the magnitude of the association differed between the two populations, the effect indicated the same correlation, supporting a potential protective role of the rs2004659 G allele across studies. By contrast, Han et al. reported a protective association for the rs9286836 G allele, but we did not observe a protective correlation for this variant in our population. Such variation across populations is common and may be related to population-level differences in genetic background, local linkage disequilibrium structure, or environmental factors that shape breast cancer risk.

*NUDT17* is on chromosome 1q21.1 and has a Nudix box that is the same as those of other NUDT family members. This gene makes a Nudix enzyme that is important for how cells break down nucleotides. NUDT17 protein hydrolyzes nucleoside triphosphates (NTPs) and deoxyribonucleoside triphosphates (dNTPs), preserving cellular nucleotide equilibrium and alleviating DNA damage resulting from metabolic disruptions. Earlier research has shown that *NUDT17* can be used to predict the outcome of clear cell renal cell carcinoma (ccRCC). In the Cancer Genome Atlas (TCGA) group, lower levels of *NUDT17* were linked to lower overall survival (OS) rates. [17]. Additional research indicates that *NUDT17* is overexpressed in breast cancer tissues and correlates with improved prognosis for patients with elevated expression levels. Studies suggest that *NUDT17* plays a critical role in initiating and progressing breast cancer [25].

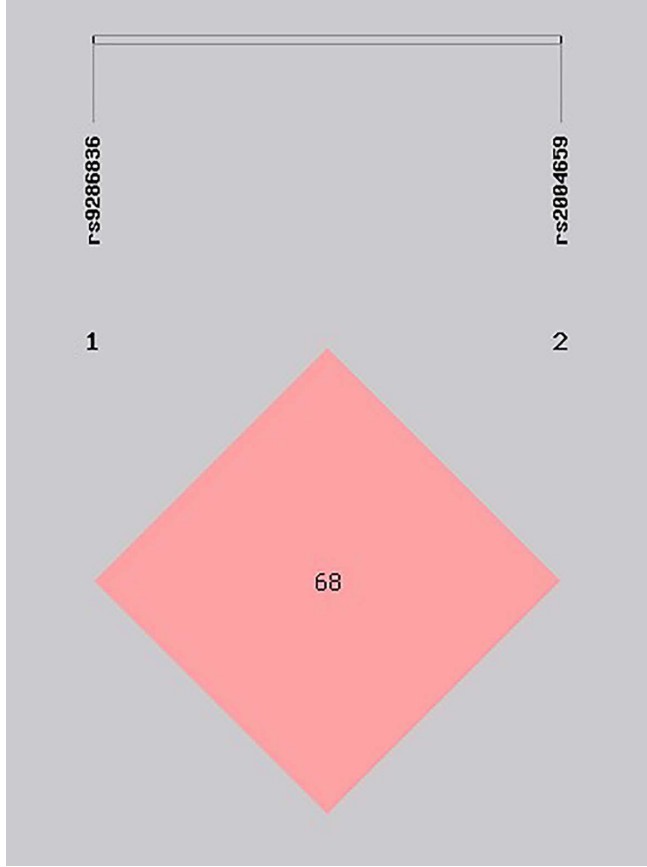

**Fig 1. Haplotype block for SNPs of the *NUDT17* rs9286836 and *NUDT17* rs2004659 polymorphisms.**

A study has shown that rs9286836, rs2004659, rs10910830, and rs10910829 are probable binding determinants associated with the regulation of *NUDT17* gene expression. Evidence suggests that rs9286836 may be linked to elevated *NUDT17* expression, thereby influencing the metabolic reprogramming of breast cancer cells [30,31]. Uncontrolled REDOX metabolism in cancer cells can lead to oxidative damage, and *NUDT17* is thought to mitigate this damage by acting on oxidized nucleotides in breast cancer [24]. Our analysis of various genetic models found no statistically significant relationship between the rs9286836 polymorphism and breast cancer risk, as indicated by p-values exceeding 0.05. Some models exhibited weak trends; however, these trends were not statistically significant and do not substantiate an association between rs9286836 and breast cancer susceptibility. Research on the Chinese population identified two protective SNPs (rs9286836 and rs2004659) and one risk SNP (rs10910830) associated with breast cancer susceptibility [29]. However, differences in genetic background and environmental exposures between the Chinese and Bangladeshi populations may influence the observed patterns of association. In the analysis of rs2004659, the AG genotype in the additive model demonstrated a 43% reduction in breast cancer development risk in contrast to the AA genotype (p = 0.007, OR = 0.57, 95% CI = 0.38 to 0.86). The dominant model (AG + GG vs. AA) indicated a 41% reduction in risk (p = 0.0098; OR = 0.59; 95% CI = 0.40 to 0.88). Similarly, the over-dominant model provided additional evidence of a protective effect, showing a 42% reduced risk for AG carriers compared to the AA + GG genotypes (p = 0.009; OR = 0.58; 95% CI = 0.39 to 0.87). The allelic model comparing the G allele to the A allele also suggested a protective effect (OR = 0.71), which was statistically significant (p = 0.037).

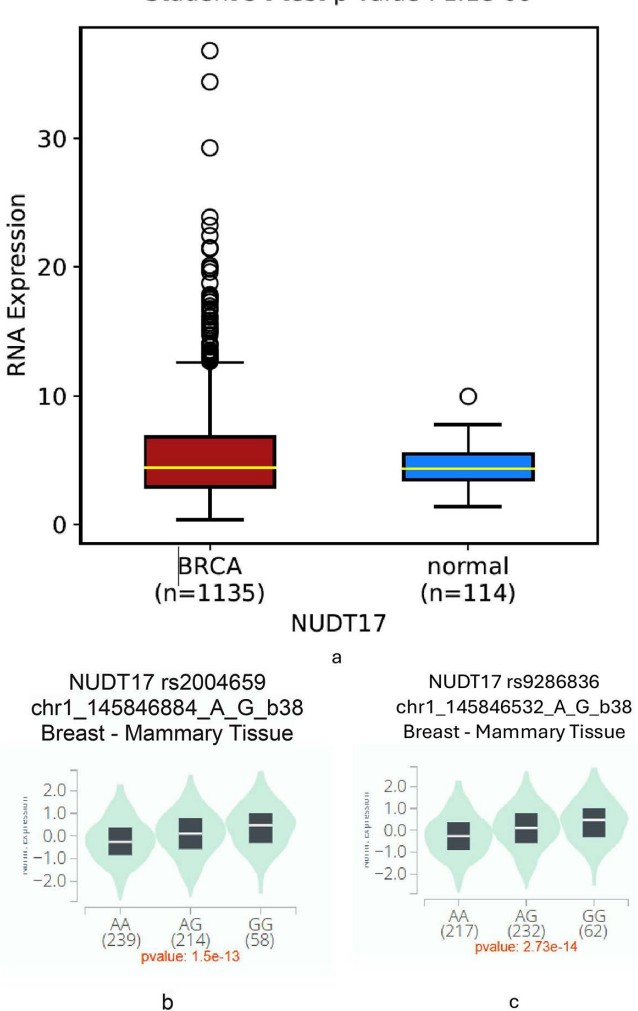

Student's t-test p-value : 1.1e-06

NUDT17

a

NUDT17 rs2004659
chr1_145846884_A_G_b38
Breast - Mammary Tissue

NUDT17 rs9286836
chr1_145846532_A_G_b38
Breast - Mammary Tissue

b

c

**Fig 2. *NUDT17* expression in breast cancer and by genotype.** 2a (OncoDB): breast cancer vs. normal tissues ($p = 1.1 \times 10^{-6}$), 2b (GTEx, rs2004659): genotype-specific expression ($p = 1.5 \times 10^{-13}$), 2c (GTEx, rs9286836): genotype-specific expression ($p = 2.73 \times 10^{-14}$).

Conversely, no association was observed for the GG homozygous genotype in the additive model (GG vs. AA, p = 0.591; OR = 0.79; 95% CI = 0.34 to 1.84) or the recessive model (GG vs. AA+AG, p = 1; OR = 1; 95% CI = 0.44 to 2.28). Earlier studies have suggested an association between the rs9286836 G allele and a reduced risk of breast cancer (p = 0.022) and that the rs2004659 G allele lowers breast cancer risk by 32% compared to the A allele (p = 0.004) [29]. Our results suggest that rs2004659 polymorphism has a protective effect on the breast cancer susceptibility. The rs9286836 GG genotype showed an increased risk, but this finding was not statistically significant. The haplotype analysis offered additional understanding by evaluating the cumulative impact of rs9286836 and rs2004659. The AA haplotype was significantly more common among patients, suggesting that it may increase susceptibility, whereas the AG and GA haplotypes appeared to be protective. This suggests that the combined effect of different alleles, rather than individual variants alone, may influence how *NUDT17* functions and how cells respond to oxidative stress. We also found a moderate degree of linkage disequilibrium between the two loci (D′ = 0.688, r² = 0.414), as shown in Fig 1, which supports the idea of a shared haplotype block. Interestingly, studies in Chinese cohorts also identified haplotypes involving these variants as influential,

with protective roles observed for certain allele combinations. The differences in haplotype effects across populations likely reflect ethnic-specific genetic architectures.The expression analyses provide functional support for the genetic findings. Data from OncoDB confirmed that *NUDT17* is upregulated in breast cancer compared with normal tissues, reinforcing its potential role in tumor biology. Moreover, GTEx demonstrated clear genotype-dependent effects for both rs2004659 and rs9286836, suggesting that these variants may influence transcriptional regulation of the gene. These observations are based on in silico datasets and therefore indicate potential functional relevance rather than experimentally validated mechanisms. Taken together, these in silico results complement our case–control analysis: while rs2004659 emerged as a protective variant at the disease level, both SNPs showed strong associations with gene expression.

## Study limitations

The main limitation of this study is the modest sample size, which may affect the precision of effect estimates for less frequent genotypes. Although 240 breast cancer cases and 240 controls were recruited, complete genotyping data were not available for all participants for every SNP. While rs9286836 was analyzed in the full cohort (240 cases and 240 controls), rs2004659 analyses were based on 204 cases and 204 controls after quality control procedures. Nevertheless, the final analyzed sample of 408 participants met the predefined requirement for 80% statistical power based on a priori calculations, although wider confidence intervals were observed for some genotype categories. In addition, functional conclusions were based on publicly available in silico data rather than laboratory-based assays. Larger multicenter studies with expanded cohorts and experimental validation are needed to confirm these associations and further define the biological relevance of NUDT17 variants in breast cancer.

## Conclusion

This study provides the first evidence that *NUDT17* rs2004659 is associated with reduced breast cancer risk in Bangladeshi women, while rs9286836 showed no significant association. These findings highlight the potential role of *NUDT17* in breast cancer susceptibility and underscore the need for larger, more diverse studies, along with functional analyses, to validate these associations and clarify their underlying biological mechanisms.

## Supporting information

**S1 Table. Socio-demographic and clinical characteristics of breast cancer patients and controls.** This table summarizes socio-demographic and selected reproductive and lifestyle characteristics of breast cancer cases and healthy controls, including religion, gap between first and second child, duration of oral contraceptive pill use, history of postmenopausal hormonal therapy, smoking status, and family history of cancer. Group comparisons were performed using appropriate statistical tests, and corresponding p-values are shown.
(DOCX)

**S1 Fig. Representative agarose gel electrophoresis (3%) showing tetra-primer ARMS-PCR genotyping of NUDT17 rs9286836.** All lanes represent individual study samples, with Lane 1 containing a 100 bp DNA ladder. Selected lanes are annotated to illustrate representative genotypes: AA genotypes show bands at 190 bp and 301 bp, AG genotypes show bands at 160 bp, 190 bp, and 301 bp, and GG genotypes show bands at 160 bp and 301 bp. Band sizes correspond to the expected amplicon lengths.
(TIF)

**S2 Fig. Representative agarose gel electrophoresis (2%) showing PCR–RFLP genotyping of NUDT17 rs2004659 following 16 h restriction enzyme digestion.** All lanes represent individual study samples, with Lane 1 containing a 100 bp DNA ladder. Selected lanes are annotated to illustrate representative genotypes: the AA genotype shows fragments

at 89 bp and 149 bp, the AG genotype shows fragments at 89 bp, 149 bp, and 238 bp, and the GG genotype shows an undigested fragment at 238 bp. Fragment sizes correspond to the expected restriction patterns.
(TIF)

**S1 Dataset. Individual-level data used for the case–control analysis. An anonymized dataset including socio-demographic variables, reproductive factors, lifestyle characteristics, and NUDT17 rs9286836 and rs2004659 genotypes for breast cancer patients and control participants.**
(XLSX)

## Acknowledgments

We sincerely thank all the participants who generously contributed their time and samples to this study. We are also grateful to the Department of Pharmacy, University of Asia Pacific, and the Department of Pharmacy, Noakhali Science and Technology University, for providing laboratory facilities and technical assistance. Our appreciation extends to the research assistants and students who supported the sample collection, genotyping, and data analysis.

## Author contributions

**Conceptualization:** Mohammad Safiqul Islam.

**Data curation:** Md Abdul Barek, Syed Masudur Rahman Dewan, Zabun Nahar, Mohammad Safiqul Islam.

**Formal analysis:** Md. Shajid Hossain Rafi, Md. Shalahuddin Millat, Mohammad Safiqul Islam.

**Funding acquisition:** Md. Shajid Hossain Rafi, Mohammad Shahriar.

**Investigation:** Md. Shajid Hossain Rafi, Md. Shalahuddin Millat.

**Methodology:** Md. Shajid Hossain Rafi, Md Abdul Barek, Mohammad Shahriar, Syed Masudur Rahman Dewan, Zabun Nahar, Mohammad Safiqul Islam.

**Resources:** Mohammad Shahriar, Syed Masudur Rahman Dewan, Zabun Nahar.

**Software:** Md. Shajid Hossain Rafi, Md. Shalahuddin Millat, Mohammad Safiqul Islam.

**Supervision:** Mohammad Safiqul Islam.

**Validation:** Md. Shalahuddin Millat, Md Abdul Barek, Syed Masudur Rahman Dewan.

**Visualization:** Md Abdul Barek, Mohammad Shahriar, Zabun Nahar.

**Writing – original draft:** Md. Shajid Hossain Rafi, Md Abdul Barek, Mohammad Safiqul Islam.

**Writing – review & editing:** Md. Shalahuddin Millat, Mohammad Shahriar, Syed Masudur Rahman Dewan, Zabun Nahar, Mohammad Safiqul Islam.

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
