## [Decision Letter · Decision Letter 0]

9 Dec 2025

Dear Dr.  Islam,

Thank you for submitting your manuscript to PLOS ONE. After careful consideration, we feel that it has merit but does not fully meet PLOS ONE’s publication criteria as it currently stands. Therefore, we invite you to submit a revised version of the manuscript that addresses the points raised during the review process.

We look forward to receiving your revised manuscript.

Kind regards,

Milad Khorasani, PhD

Academic Editor

PLOS One

Journal Requirements:

“The study was partially financially supported by the Department of Pharmacy at the University of Asia Pacific.”

4. We note that there is identifying data in the Supporting Information file < Supplementary materials.rar>. Due to the inclusion of these potentially identifying data, we have removed this file from your file inventory. Prior to sharing human research participant data, authors should consult with an ethics committee to ensure data are shared in accordance with participant consent and all applicable local laws.

-Location data

Reviewer's Responses to Questions

**Comments to the Author**

1. Is the manuscript technically sound, and do the data support the conclusions?

Reviewer #1: Yes

Reviewer #2: Yes

Reviewer #3: Yes

2. Has the statistical analysis been performed appropriately and rigorously?

Reviewer #1: No

Reviewer #2: No

Reviewer #3: Yes

3. Have the authors made all data underlying the findings in their manuscript fully available?

Reviewer #1: Yes

Reviewer #2: Yes

Reviewer #3: Yes

4. Is the manuscript presented in an intelligible fashion and written in standard English?

Reviewer #1: Yes

Reviewer #2: Yes

Reviewer #3: Yes

Reviewer #1: 1. General Comments

The study is well-designed with comprehensive data collection and appropriate analysis. The study provides valuable preliminary data on breast cancer genetic susceptibility in Bangladeshi women. However, several sections need some clarification and improvements.

2. Specific Comments

Abstract

• This section is clearly described but needs to avoid excessive inclusion of numerical values from results and rather focus on significant findings.

• Try and narrative results for logistic regression then long OR/CIs numbers.

Introduction

• This section is detailed and backed with relevant literature. As for the section of NUDIX gene, the authors must focus on its role specifically with regards to breast cancer.

• Need to strengthen the justification for selecting rs9286836 and rs2004659 SNPs and emphasize the need for this analysis in Bangladeshi women.

Materials and Methods

Methods are well-detailed however, need to clarify;

• Selection criteria for study subjects and whether cases/controls were matched.

• Include genotype call validation (e.g., random re-genotyping for QC).

• In the sample size estimation paragraph do include actual sample sizes.

• In statistical analysis do indicate how Hardy–Weinberg equilibrium was tested. Also clarify Bonferroni correction (pre-specified or post hoc). If possible do consider adjusting logistic regression models adjusted for potential confounders (e.g., age, BMI, reproductive factors) for stronger outcomes.

Results

• Overall results are well organized but could be meaningful by summarizing them with relevant details.

• Data analysis presented in tables is comprehensive and well structured but needs to highlight statistically significant results e.g. of rs2004659 SNP and avoid overinterpretation for rs9286836 SNP.

• Gel electrophoresis figures are more suited to the supplementary section.

Discussion

• The results interpretation is well aligned and descriptive but avoid emphasis on some of the non-significant results and speculative statements.

• The functional implications of rs2004659 results need to be carefully discussed as the analyses are in silico and not experimentally validated.

• Inclusion of a study limitations paragraph will strengthen the manuscript.

• It’s worth comparing the Bangladeshi population with the Chinese cohort but do focus on the potential genetic variability of both ethnic groups.

• In the conclusion section, avoid repeating results and focus on the main findings.

Language and Style

• The English is fluent and formal, but a few sentences are long and could be simplified.

• Maintain consistent use of “breast cancer” instead of abbreviations (BC) unless used frequently in tables.

• Check punctuation, spacing, and consistent decimal formatting (e.g., “p = 0.0098” rather than “p=0.0098”).

• Ensure consistent use of scientific italics for gene symbols (NUDT17).

Figures and Tables

• Figures quality could be improved by enhancing resolution and clearer labelling.

• The rarely used sociodemographic variables presented in table 2 could be moved to Supplementary materials.

References

• All references are properly selected and up-to-date.

• Authors must ensure PLOS ONE style formatting.

• Citing papers on role of NUDIX enzymes to oxidative stress response in cancer will be a plus.

Reviewer #2: PONE-D-24-22063R1

Association of Vitamin D levels and Vitamin D receptor gene polymorphism with obesity in Bangladeshi school-going children: A cross-sectional study

PLOS ONE

My comments:

This is a very interesting study which will attract the attention of scientists and readers globally. However, manuscript requires a major revision and can be considered for publication after addressing the following comments.

1. In abstract authors require to add this sentence “An association between the NUDT17 polymorphisms rs9286836 and rs2004659 and the risk of breast cancer in Bangladeshi Women has not been studied previously and thus, this study aimed to determine this correlation. It will clarify the novelty of the study.

2. How was the number of BC participants determined? Sample size is small and should be at least 400 or more to be statistically valid according to the existing number of BC patients in Bangladesh. Please justify.

3. There are four SNPs; rs9286836, rs2004659, rs10910830, and rs10910829 probable binding determinants associated with the regulation of NUDT17 gene expression. Why only two SNPs rs9286836, rs2004659 were studied? I would recommend investigating the association of rs10910830, and rs10910829 with BC risk as well.

4. Genotyping was performed using tetra-primer ARMS-PCR for rs9286836 and PCR–RFLP for rs2004659, why two different techniques were used? Please explain.

5. Hardy-Weinberg equilibrium should be determined for allelic and genotypic model of each SNP. Include this in the statistical evaluation section and its findings in the result section.

6. In the introduction, please include the study conducted in Chinese women by Han et all., 2024 to reveal this association as a first line of the last paragraph.

7. In the discussion, include detail on allelic association with BC risk from the study conducted by Han et all, 2024 (Title of the study: Association between NUDT17 polymorphisms and breast cancer risk) and compare your study findings with their findings.

8. In the discussion section, please include references for this statement “Previous studies on the NUDT17 rs9286836 and rs2004659 polymorphisms suggested a possible correlation between these genetic variants and cancer susceptibility”.

9. In the in the first line of conclusion, please replace “individuals” by “women”.

Reviewer #3: In introduction section:

1. Please add the background information regarding the association of SNPs of interest with breast cancer seen in other ethnicity (if available).

2. Avoid using the information regarding the current study method in the last part of the introduction, shift them to methodology. State the objective only.

Methodology:

1. Why the own study data was not used during in silico gene expression analysis?

Result:

1. It is better to describe as 'no association' for rs9286836 in stead of mentioning 'associated but not statistically significant'.

Overall:

1. Please write the full form when first using the acronyms/abbreviations.

2. I wonder why you have no co-authors from NICRH. How did you manage sample and data collection without collaborating with them?

**Do you want your identity to be public for this peer review?** For information about this choice, including consent withdrawal, please see our Privacy Policy

Reviewer #1: No

Reviewer #2: **Yes:** Dr. Raushanara Akter

Reviewer #3: **Yes:** Mashfiqul Hasan

---

## [Author Response · Author response to Decision Letter 1]

9 Feb 2026

PONE-D-25-57284

Association of NUDT17 rs9286836 and rs2004659 Variants with Breast Cancer Risk in Bangladeshi Women

PLOS One

Editor:

“The study was partially financially supported by the Department of Pharmacy at the University of Asia Pacific.”

4. We note that there is identifying data in the Supporting Information file < Supplementary materials.rar>. Due to the inclusion of these potentially identifying data, we have removed this file from your file inventory. Prior to sharing human research participant data, authors should consult with an ethics committee to ensure data are shared in accordance with participant consent and all applicable local laws.

-Location data

Response:

The revised manuscript has been prepared in accordance with the Editor's and reviewers' suggestions. The patients’ and controls' IDs have been removed from the ‘Research data’ file and provided with just numbering as patient and control IDs. Funder information has been moved to the cover letter.

Reviewer's Responses to Questions

Comments to the Author

1. Is the manuscript technically sound, and do the data support the conclusions?

Reviewer #1: Yes

Reviewer #2: Yes

Reviewer #3: Yes

2. Has the statistical analysis been performed appropriately and rigorously?

Reviewer #1: No

Reviewer #2: No

Reviewer #3: Yes

3. Have the authors made all data underlying the findings in their manuscript fully available?

Reviewer #1: Yes

Reviewer #2: Yes

Reviewer #3: Yes

4. Is the manuscript presented in an intelligible fashion and written in standard English?

Reviewer #1: Yes

Reviewer #2: Yes

Reviewer #3: Yes

5. Review Comments to the Author

Reviewer #1: 1. General Comments

The study is well-designed with comprehensive data collection and appropriate analysis. The study provides valuable preliminary data on breast cancer genetic susceptibility in Bangladeshi women. However, several sections need some clarification and improvements.

Response:

We appreciate the positive evaluation. Following the reviewer’s suggestions, we have revised the Abstract, Introduction, Methods, Results, Discussion, and presentation of figures and tables to improve clarity, focus, and compliance with PLOS ONE standards.

2. Specific Comments

Abstract

• This section is clearly described but needs to avoid excessive inclusion of numerical values from results and rather focus on significant findings.

• Try and narrative results for logistic regression then long OR/CIs numbers.

Response:

Thank you for this comment. We have revised the abstract to reduce the emphasis on detailed numerical values and instead focus on the main findings that are most relevant biologically and clinically. The results are now described in a narrative manner, particularly for the logistic regression analyses, with emphasis placed on the statistically significant protective effect of the rs2004659 variant, rather than listing multiple odds ratios and confidence intervals.

3. Introduction

• This section is detailed and backed with relevant literature. As for the section of NUDIX gene, the authors must focus on its role specifically with regards to breast cancer.

• Need to strengthen the justification for selecting rs9286836 and rs2004659 SNPs and emphasize the need for this analysis in Bangladeshi women.

Response:

In addition, the introduction has been refined to better contextualize the role of the NUDT17 gene specifically in breast cancer. We have strengthened the rationale for selecting rs9286836 and rs2004659 by highlighting prior evidence, their potential functional relevance, and the lack of population-specific data. The need to evaluate these variants in Bangladeshi women is now more clearly stated, given the distinct genetic background and the limited existing data from this population.

4. Materials and Methods

Methods are well-detailed however, need to clarify;

• Selection criteria for study subjects and whether cases/controls were matched.

• Include genotype call validation (e.g., random re-genotyping for QC).

• In the sample size estimation paragraph do include actual sample sizes.

• In statistical analysis do indicate how Hardy–Weinberg equilibrium was tested. Also clarify Bonferroni correction (pre-specified or post hoc). If possible do consider adjusting logistic regression models adjusted for potential confounders (e.g., age, BMI, reproductive factors) for stronger outcomes.

Response:

We thank the reviewer for these valuable suggestions. We clarified the inclusion/exclusion criteria and explicitly stated age, sex, and BMI matching between cases and controls. Specifically, we have clarified the selection criteria and matching strategy for cases and controls, added a description of genotype call validation for quality control, explicitly stated the actual sample size used in the a priori power calculation, and expanded the statistical analysis section to describe Hardy-Weinberg equilibrium testing and the pre-specified use of Bonferroni correction. We also clarified the rationale for reporting crude logistic regression models and addressed potential confounding.

5. Results

• Overall results are well organized but could be meaningful by summarizing them with relevant details.

• Data analysis presented in tables is comprehensive and well structured but needs to highlight statistically significant results e.g. of rs2004659 SNP and avoid overinterpretation for rs9286836 SNP.

• Gel electrophoresis figures are more suited to the supplementary section.

Response:

We appreciate the reviewer's constructive suggestions. The results section has been revised to provide a clearer summary of the major findings, with emphasis on the statistically significant association observed for rs2004659 and careful wording for the non-significant result for rs9286836. Interpretations for rs9286836 have been limited to descriptive statements without speculative inference. The gel electrophoresis figures (Figures 1 and 2) have been moved to the Supplementary Information (S1 Fig and S2 Fig) as recommended.

6. Discussion

• The results interpretation is well aligned and descriptive but avoid emphasis on some of the non-significant results and speculative statements.

• The functional implications of rs2004659 results need to be carefully discussed as the analyses are in silico and not experimentally validated.

• Inclusion of a study limitations paragraph will strengthen the manuscript.

• It’s worth comparing the Bangladeshi population with the Chinese cohort but do focus on the potential genetic variability of both ethnic groups.

Response:

Thank you for these comments. We have gone through the discussion and made several small but important changes. We reduced the attention given to rs9286836, where the results were not significant, and removed wording that could be taken as speculative.

We also made it clearer that the functional interpretation of rs2004659 comes from in silico expression data and has not yet been tested experimentally. A separate paragraph on study limitations was added to explain the constraints of the sample size and the use of public databases.

The comparison with Chinese studies was revised as well, so that it now reflects possible differences between the two populations rather than assuming the same genetic effects. These changes were made to keep the discussion accurate and appropriately cautious.

7. In the conclusion section, avoid repeating results and focus on the main findings.

Response:

The conclusion section has been modified according to the reviewer’s suggestion.

8. Language and Style

• The English is fluent and formal, but a few sentences are long and could be simplified.

•Maintain consistent use of “breast cancer” instead of abbreviations (BC) unless used frequently in tables.

• Check punctuation, spacing, and consistent decimal formatting (e.g., “p = 0.0098” rather than “p=0.0098”).

• Ensure consistent use of scientific italics for gene symbols (NUDT17).

Response

We carefully reviewed the entire manuscript to improve readability and consistency. Several long or complex sentences were simplified without altering their scientific meaning. The term “breast cancer” is now used consistently throughout the text, with abbreviations avoided except where necessary in tables. Punctuation, spacing, and decimal formatting were standardized across the manuscript (e.g., p = 0.0098), and gene symbols, including NUDT17, were checked to ensure consistent use of scientific italics.

9. Figures and Tables

• Figures quality could be improved by enhancing resolution and clearer labelling.

• The rarely used sociodemographic variables presented in table 2 could be moved to Supplementary materials.

Response:

All figures were revised to improve resolution and visual clarity, with clearer labels and legends to enhance interpretability. Additionally, these figures have now been moved to the supplementary materials, as per your previous suggestion. In response to the suggestion regarding Table 2, sociodemographic variables that were not directly relevant to the genetic association analyses were moved to the Supplementary Materials (S1Table), allowing the main tables to focus on variables central to the study objectives.

10. References

• All references are properly selected and up-to-date.

• Authors must ensure PLOS ONE style formatting.

• Citing papers on role of NUDIX enzymes to oxidative stress response in cancer will be a plus.

Response:

In response to the reviewer’s suggestion, the manuscript already cites key studies describing the role of NUDIX family enzymes in oxidative stress and nucleotide sanitization, including foundational biochemical work on MTH1/MTH2 and oxidized nucleotide metabolism [24,30,31], as well as recent evidence linking NUDT17 polymorphisms to breast cancer susceptibility [25].

Reviewer #2: PONE-D-24-22063R1

Association of Vitamin D levels and Vitamin D receptor gene polymorphism with obesity in Bangladeshi school-going children: A cross-sectional study

PLOS ONE

My comments:

This is a very interesting study which will attract the attention of scientists and readers globally. However, manuscript requires a major revision and can be considered for publication after addressing the following comments.

Response:

Thank you very much for your comments. Although the title is not the title of our manuscript, all the comments are relevant to our manuscript.

We sincerely thank the reviewer for the constructive and detailed comments. Several of the points raised by you (Reviewer #2) overlap with those raised by Reviewer #1 and have already been addressed in the revised manuscript with tracked changes. For clarity, we respond to each comment individually below and indicate where revisions were made.

1. In abstract authors require to add this sentence “An association between the NUDT17 polymorphisms rs9286836 and rs2004659 and the risk of breast cancer in Bangladeshi Women has not been studied previously and thus, this study aimed to determine this correlation. It will clarify the novelty of the study.

Response:

We agree wi

---

## [Decision Letter · Decision Letter 1]

23 Feb 2026

Association of NUDT17 rs9286836 and rs2004659 Variants with Breast Cancer Risk in Bangladeshi Women

PONE-D-25-57284R1

Dear Dr. Safiqul Islam,

We’re pleased to inform you that your manuscript has been judged scientifically suitable for publication and will be formally accepted for publication once it meets all outstanding technical requirements.

Kind regards,

Milad Khorasani, PhD

Academic Editor

PLOS One

Additional Editor Comments (optional):

Reviewers' comments:

Reviewer's Responses to Questions

**Comments to the Author**

Reviewer #1: All comments have been addressed

Reviewer #2: All comments have been addressed

2. Is the manuscript technically sound, and do the data support the conclusions?

Reviewer #1: Yes

Reviewer #2: Yes

3. Has the statistical analysis been performed appropriately and rigorously?

Reviewer #1: Yes

Reviewer #2: Yes

4. Have the authors made all data underlying the findings in their manuscript fully available?

Reviewer #1: Yes

Reviewer #2: Yes

5. Is the manuscript presented in an intelligible fashion and written in standard English?

Reviewer #1: Yes

Reviewer #2: Yes

Reviewer #1: The authors have tried to address all the comments/suggestions raised by the reviewers. The current version of the revised manuscript looks much improved.

Reviewer #2: (No Response)

**Do you want your identity to be public for this peer review?** For information about this choice, including consent withdrawal, please see our Privacy Policy

Reviewer #1: No

Reviewer #2: **Yes:** Dr. Raushanara Akter

---

## [Editor Report · Acceptance letter]

PONE-D-25-57284R1

PLOS One

Dear Dr. Islam,

I'm pleased to inform you that your manuscript has been deemed suitable for publication in PLOS One. Congratulations! Your manuscript is now being handed over to our production team.

Kind regards,

on behalf of

Dr. Milad Khorasani

Academic Editor

PLOS One